# Development of a Variable-Frequency Hammering Method Using Acoustic Features for Damage-Type Identification

**Xi Huang [1], Huang Huang [1] and Zhishen Wu [2,\*]**

1   Department of Urban and Civil Engineering, Ibaraki University, Nakanarusawa-Cho 4-12-1,
    Hitachi 316-8511, Japan
2   International Institute for Urban Systems Engineering, Southeast University, Nanjing 211189, China
\*   Correspondence: zswu@seu.edu.cn; Tel.: +86-25-52091294

**Abstract:** Hammer testing, a nondestructive testing method, has been demonstrated to provide information on structural damage. One of the biggest challenges with this testing method is the simultaneous identification of surface, internal, and composite damage (consisting of both surface and internal damage) in a complex environment, such as post-disaster. A method of identification based on variable-frequency hammering is proposed to solve this problem. The importance and feasibility of using variable-frequency impact hammers and the generated acoustic data to identify multiple types of damage in concrete structures are presented. First, a type of variable-frequency hammering acoustic feature was generated using acoustic feature extraction and selection based on the acoustic data obtained from variable-frequency hammering. Second, a damage recognition model was established using a support vector machine to identify four types of damage occurring simultaneously in the same concrete member specimens, including a type of composite damage with two types of damage occurring simultaneously within 20 mm. Finally, the feasibility of this variable-frequency hammering method was verified experimentally. This method exhibited good performance, with an accuracy of 97.8%; moreover, the method ensures that the feature dimensionality remains unchanged while increasing the effective information of the data.

**Keywords:** hammering test; nondestructive testing; acoustic data; machine learning; feature selection

## 1. Introduction

The detection of concrete, particularly with high precision, is essential for social infrastructures such as bridges and tunnels that are easily affected by disasters and ageing [1]. In recent years, many studies have been conducted on the detection of aging infrastructure, and new technologies, such as sensors and ultrasound, have been used to process complex information, save labor, and reduce costs [2,3]. However, some problems remain, such as high cost and heavy weight. The hammering test is a traditional method of inspection that is very effective and is widely used for the maintenance of bridges and tunnels. However, with the current shortage of skilled professionals, it is becoming increasingly difficult to rely solely on human sensory inspection methods for the prevention of bridge and tunnel deterioration accidents and the performance of large-scale, mobile and rapid detection of post-disaster infrastructure damage.

To date, a tremendous amount of research has been conducted on damage detection and hammering tests. In the past, time series methods of damage detection based on Autoregressive Parameters, Intrinsic Mode Function and Hilbert Spectrum, and Signal Components were widely used [4]. Gillich et al. performed modal identification and damage detection in time–frequency analysis using an algorithm based on time–frequency analysis and power spectrum [5]. Avci et al. reviewed the application of traditional methods of machine and deep learning in damage detection, demonstrating that ML-based models can be applied in damage detection [6]. In the past, hammering test applications

have included the diagnosis of concrete structures [7–11], such as diagnosis systems for spalling tiles on the facades of high-rise buildings [12] and the construction of cavity and crack detection algorithms [13]. However, these applications typically limit the number of instances of damage within the detection area to a specific number. Furthermore, they consider only a single type of damage within the target detection area for each hammering point.

However, under the influence of disasters and aging, the inspection environment becomes complex. Moreover, there are typically different types of damage in the target area, and detection is made difficult by the varying extent of their damage; for example, internal and surface cracks may be located in the target detection area of a hammering point. In this case, the simultaneous identification of the type, size, and depth of the damage means that the used qualitative and quantitative information is in the same dimension, which leads to an increased possibility that this will be misjudged.

To obtain damage information, most existing studies have focused on acoustic data obtained from single-frequency hammering, lacking the means to increase the damage information obtained from the hammering test method. This study proposes a method of acoustic concrete damage detection based on variable-frequency hammering, which is different from the usual single-frequency hammering method, to achieve the classification of multiple categories of damage occurring in the same concrete member. The acoustic data obtained from variable-frequency hammering are combined with feature selection and feature extraction to enhance the effective usage of the data without increasing its dimensionality. Finally, a detection model is established using a support vector machine (SVM), and complex multi-class damage is successfully identified. This research focuses on the innovation of the hammering method. In fact, the ML model is used as the carrier to improve the hammering detection method itself. This provides a new and promising method for hammering detection.

## 2. Principle of Variable-Frequency Hammering

Acoustic investigations of hammering tests have often concentrated on the analysis of single-frequency hammering's acoustic features [14], determining damage on the basis of variations in the excitation effect obtained by hammering the detection area. The excitation effect obtained by hammering the detection area is reflected acoustically using the power spectral density (PSD) [15], as depicted in Figure 1, which is a PSD schematic diagram of single-frequency and variable-frequency hammering to determine the excitation effect of the detection area.

During the study of hammering detection, we discovered that the excitation effect of the damage area differs for different hammering frequencies when the detection area contains just one type of damage, as observed on the left side of Figure 1. Furthermore, the highest power, which corresponds to the hammering frequency at which the shaded portion is the largest, is also the optimal excitation frequency. The effective information gleaned from a single-frequency hammering; however, is insufficient to discern between the various types of damage when more than one type of damage is present in a hammering area.

Unlike with the traditional hammering method, the proposed variable-frequency hammering method does not start with acoustic processing, which improves the useful information that can be used to identify the damage by activating the detection area with various hammering frequencies. On the right-hand side of Figure 1, a schematic representation of the acoustic data obtained using the variable frequency hammering method is illustrated. To identify more intricate damages, the excitation effect produced by altering the hammering frequency allows the highest number of shaded sections to be obtained. This provides the biggest informational contrast with other detection areas.

Therefore, an identification method based on variable-frequency hammering is proposed in this study. As shown in Figure 2, for the existing single-frequency hammering method, the acoustic data obtained from the single-frequency hammering and extracted

acoustic features were used for machine learning. In this study, the acoustic data of multiple hammer frequencies obtained with variable-frequency hammering were used to increase the effective information in the vertical direction, as shown in Figure 2. The acoustic features were obtained by acoustic feature extraction, and the dimension was then reduced to the same dimension as that of the single-frequency hammering through feature selection; finally, the selected features were used for machine learning. This study aims to increase the valid acoustic information for concrete damage classification without increasing the acoustic feature dimensionality or causing a dimensional disaster.

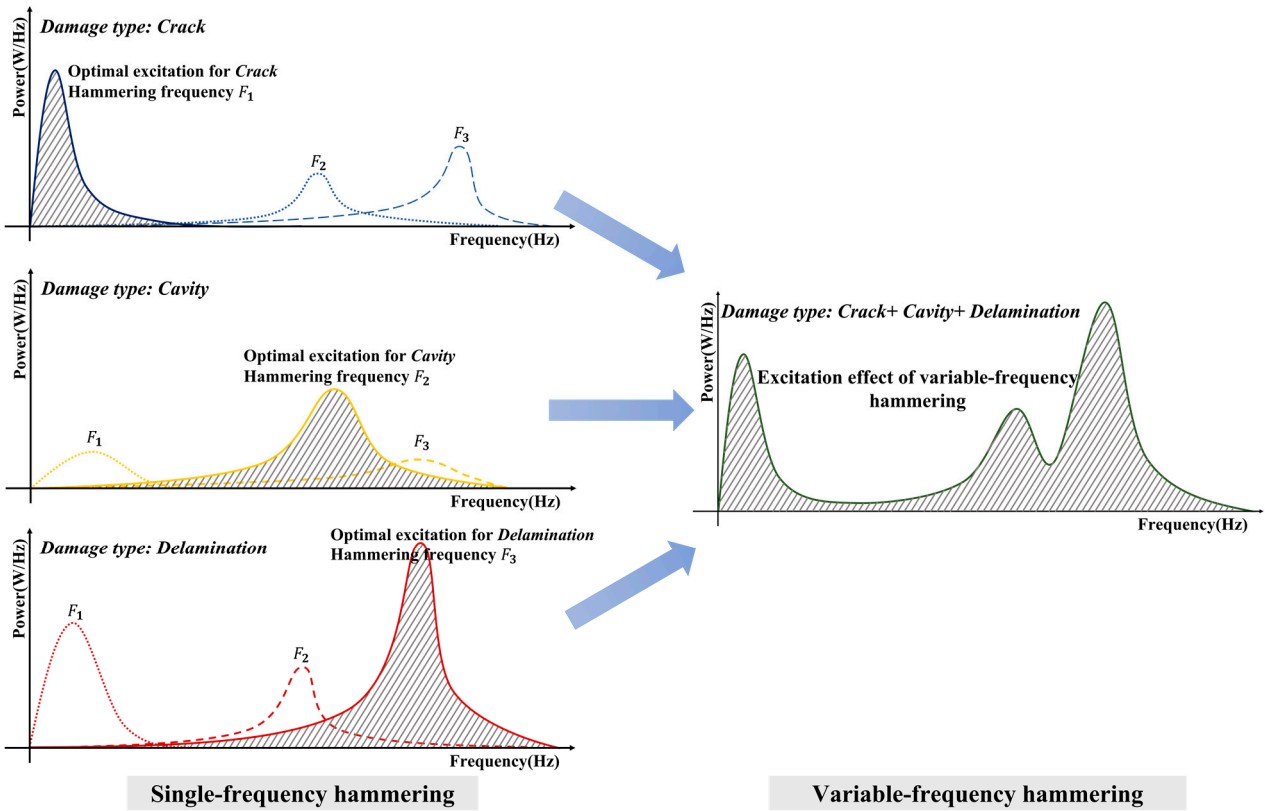

**Figure 1.** Schematic diagram of the excitation effect under different hammering methods.

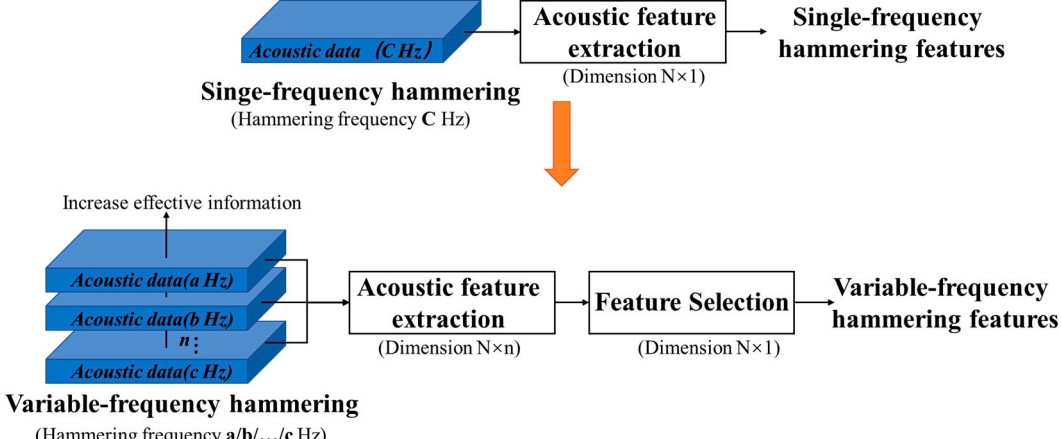

**Figure 2.** Diagram of the difference between variable-frequency hammering and single-frequency hammering.

The flowchart of the proposed method is shown in Figure 3. After recording the acoustic signal using a microphone, it was divided into two processes: training and testing. The first process is the training process, including the use of Mel-frequency cepstrum coefficients (MFCCs) were used for feature extraction of acoustic signals obtained from single-frequency and variable-frequency hammering and feature selection of MFCCs for variable-frequency hammering. Subsequently, SVMs were used to model the MFCCs for single-frequency and variable-frequency hammering. During the testing process, the MFCCs of single-frequency and variable-frequency hammering after feature selection were matched with those of the established model. In this study, two different models were established and compared based on single-frequency and variable-frequency hammering.

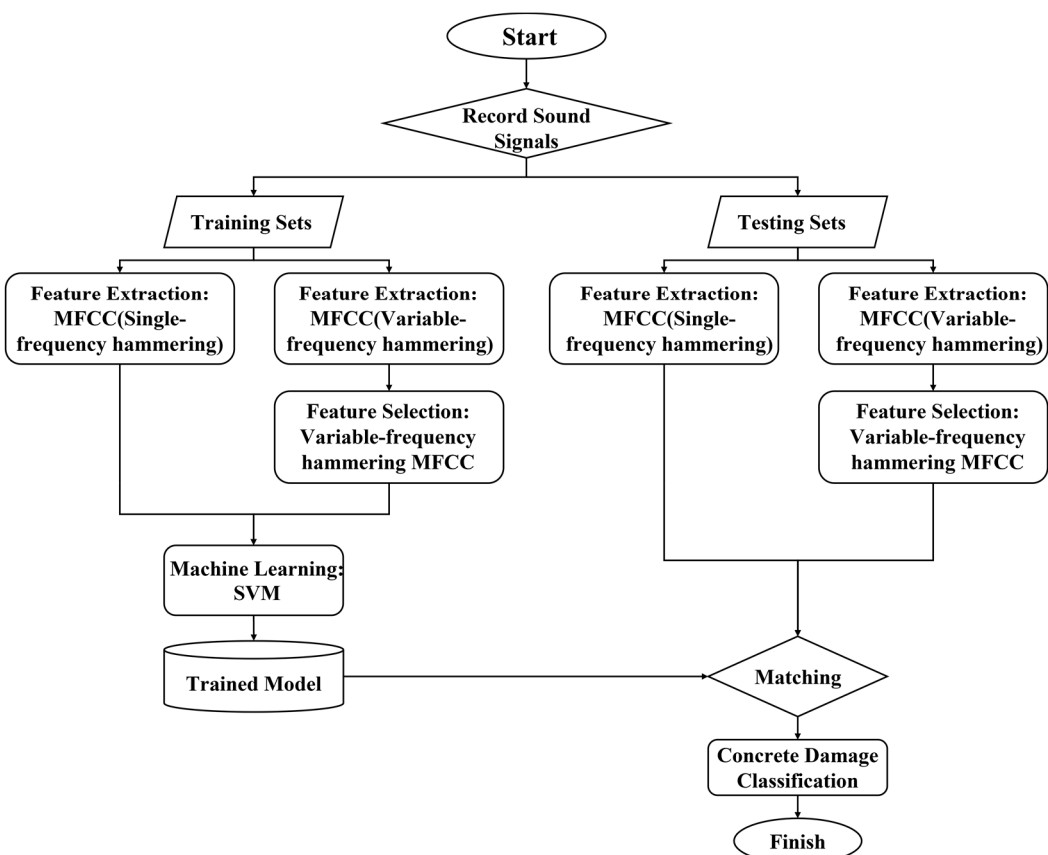

**Figure 3.** Flowchart of the detection process.

## 3. Theoretical Background

### 3.1. Relief Algorithm

Kira et al. proposed Relief in 1992 [16]. This method borrows the idea of the nearest neighbor learning algorithm. Its theoretical basis is that a good feature should make the eigenvalues of the same type of samples in the nearest neighbor the same or similar. However, the values of different samples of the nearest neighbor are different or very different. Accordingly, each feature is assigned a corresponding weight to sort it. The greater the weight of the feature, the stronger its classification ability. In contrast, lower weight indicates that the classification ability of the features is weaker. The corresponding feature selection can be performed by setting the threshold of the feature weight or the number of feature subsets.

However, Relief is effective only when the training samples are divided into two categories. Kononenko extended the Relief algorithm to the ReliefF algorithm, which can be applied to multi-class sample cases [17,18]. When dealing with multi-class problems, the algorithm selects the nearest neighbor sample from each different class sample instead of

selecting from all different samples. We select $k$ nearest neighbor samples, take the average value to obtain each feature weight, and then obtain the correlation of each feature in each sample instance with the class. We can then set a threshold to determine whether the features are valid or invalid or select the features with the largest m' weights and remove other features.

The ReliefF algorithm is described in Algorithm 1 ReliefF.

---

**Algorithm 1** ReliefF

---

Input: Dataset $D$

        Number of examples to be selected by the algorithm $m$
        Number of nearest neighbors $k$

---

Output: Vector of feature importance values $W$

---

1:      Initialize: set all weight $W[A_l] = 0$, $l = 1, 2, \ldots, L–1, L$
2:      **for** $i = 1$ **to** $m$ **do**
3:        randomly select an instance $R_i$ from $D$
4:        find $k$ nearest hits $H_j$ of $R_i$
5:        **for** each class $C \neq$ class $(R_i)$ **do**
6:          from class $C$ find k nearest misses $M_j(c)$ of $R_i$
7:        **end for**
8:        **for** $l = 1$ **to** $L$ **do**

9:
$$W(A_l) = W(A_l) - \sum_{j=1}^{k} diff\left(A_l, R_i, H_j\right) / (m \cdot k)$$
$$+ \sum_{C \neq class(R_i)} \left[ \frac{p(c)}{1-p(class(R_i))} \sum_{j=1}^{k} diff\left(A_l, R_i, M_j(C)\right) \right] / (m \cdot k)$$

10:     **end for**
11:    **end for**

---

In the algorithm, $H$ represents the nearest neighbor sample of the same type as $R$, $M$ represents the nearest neighbor sample of the same type as $R$, $p(c)$ represents the distribution probability of the class, and the function $diff$ is used to calculate the difference between the features of two different samples $Instance_1$ and $Instance_2$ as follows.

For discrete features:

$$diff\left(A_l, R_i, H_j\right) = \begin{cases} 0; & value(A, I_1) = value(A, I_2) \\ 1; & otherwise \end{cases} \tag{1}$$

For continuous features:

$$diff\left(A_l, R_i, H_j\right) = \frac{value(A, I_1) - value(A, I_2)}{\max(A) - \min(A)} \tag{2}$$

Here, $I_1$ and $I_2$ are two samples, and $value(A, I_1)$ refers to the value of the $A$th feature of sample $I_1$. After finding the relevance weights $W$ of each feature and class, they are sorted; the features whose relevance is greater than a certain threshold value constitute the final feature subset such that invalid features are eliminated.

### 3.2. Support Vector Machine Method

Vapnik et al. (2013) proposed an SVM based on the structural risk-minimization principle in statistical learning theory [19]. The SVM can maximize the generalization ability of the learning machine by obtaining small errors in the discriminant function obtained from limited datasets, even for independent test sets. Furthermore, SVM is a convex quadratic optimization problem that guarantees that the extreme solution found is the global optimal solution. Research on SVMs can be traced to the end of the 1970s, and SVMs have been successfully applied to classification and regression problems in nondestructive testing and structural health monitoring, including structural damage

detection [20–22], dam safety prediction [23], eddy vibration response prediction [24], and impact detection and localization [25].

As shown in Figure 4, assume two types of linearly separable training sample sets $\{(\varkappa_j, y_i), i = 1, 2, 3, \ldots, n\}$, where $x_i \in R^m$, $m$ is the $i$th training sample, and $y_i \in \{-1, 1\}$ is the label of the sample category. The general form of the linear discriminant function is

$$f(x) = w \cdot x + b, \tag{3}$$

where $w$ is the normal vector of the optimal classification hyperplane, $b$ is a constant, and the corresponding classification surface equation is $w \cdot x + b = 0$. The discriminant function is normalized such that all samples of the two categories satisfy $f(x) \geq 1$. At this time, the sample closest to the classification surface is $f(x) = 1$, which requires that all samples of the classification surface be classified correctly, that is, satisfy

$$y_i = [(w \cdot x_i) + b] - 1 \geq 0, \; i = 1, \ldots, n. \tag{4}$$

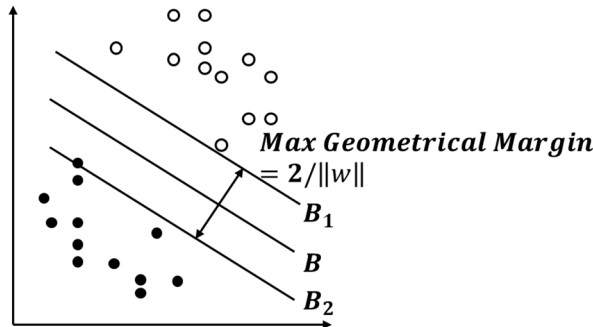

**Figure 4.** Support vectors and optimal hyperplane of a linearly separable case.

Thus, the classification interval is equal to $2/\|w\|$, and the largest interval is equivalent to the smallest value of $\|w^2\|$. A classification surface that satisfies Equation (3) and minimizes $\|w^2\|$ is the optimal classification line $B$ shown in Figure 4, where $\|\cdot\|$ represents the Euclidean distance. Between the two types of data samples, the samples closest to the classification surface and the data samples on hyperplanes $B_1$ and $B_2$ parallel to the classification surface $B$ are the data samples that cause the sign in Equation (3) to be equal; these samples are support vectors.

When the two types of samples are linearly indistinguishable, the idea of the non-linear discrimination problem is to first map the input vector into a high-dimensional space by a non-linear transformation, and then perform classification operations in this high-dimensional feature space to obtain the optimal classification surface.

*3.3. Mel-Frequency Cepstrum Coefficients Method*

As valid acoustic features, MFCCs are widely used in civil engineering [26–31]. The MFCCs represent the cosine transformation result of the real logarithm of the short-term energy spectrum on a frequency scale. The relationship between the frequencies on the Mel frequency and Hertzian scales is as follows:

$$Mel(f) = 2595 \cdot \lg(1 + \frac{f}{700}) \tag{5}$$

Figure 5 shows the general process for determining MFCCs. There are four steps: preprocessing, fast Fourier transform (FFT), Meier frequency filtering, and discrete cosine transform (DCT).

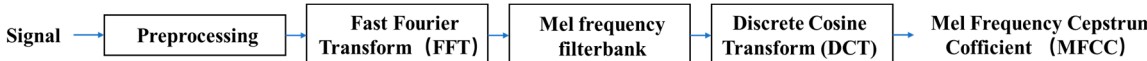

**Figure 5.** Flowchart of MFCCs.

Step 1: The preprocessing step is divided into two parts: frame splitting and windowing. Frame splitting can be completed by continuous segmentation; however, the overlapping segmentation method is generally used to make the transition between frames smooth and consistent; that is, the end of each frame overlaps with the head of the next frame. We need to multiply the speech frame by the window function to reduce the truncation effect of the speech frame and reduce the slope of the two ends of the frame so that the two ends of the speech frame do not cause sharp changes and smooth transition to 0. Let the frame signal be $x(n)$ and the window function be $w(n)$. Then, signal $y(n)$ after adding the window is

$$y(n) = x(n) \cdot w(n), \tag{6}$$

where $0 \leq n \leq N - 1$; $N$ is the number of sampling points for each frame. The most commonly used window function is the Hamming window.

$$w(n) = 0.54 - 0.46cos\frac{2\pi n}{N-1} \quad 0 \leq n \leq N - 1 \tag{7}$$

Step 2: Because discrete Fourier transform involves a large amount of calculation, an efficient FFT can be used to transform the speech frame from the time domain to the frequency domain.

Step 3: The discrete spectrum obtained from the FFT is filtered using a sequential triangular filter to obtain a set of coefficients $m_1, m_2, \ldots$. The center frequency $f(i)$ of each triangular filter in this filter set is equally spaced on the Mel frequency axis, and its span is also equal on the Mel frequency scale. The number of filters $p$ is determined by the cutoff frequency of the signal, and all filters cover, in general, from 0 Hz to the Nyquist frequency, that is, one-half of the sampling rate.

The equation for calculating $m_i$ is as follows:

$$m_i = \ln\left(\sum_{k=0}^{N-1} |x(k)| \cdot H_i(k)\right) \quad i = 1, 2, \ldots, p \tag{8}$$

The calculation of $H_i(k)$ is carried out as follows.

$$H_i(k) = \begin{cases} 0 & k < f[i-1] \ or \ k > f[i+1] \\ \frac{2(k-f[i-1])}{(f[i+1]-f[i-1])\cdot(f[i]-f[i-1])} & f[i-1] \leq k \leq f[i] \\ \frac{2(f[i+1]-k)}{(f[i+1]-f[i-1])\cdot(f[i+1]-f[i])} & f[i] \leq k \leq f[i+1] \end{cases} \tag{9}$$

Here, $y[i]$ is the center frequency of the triangular filter, which satisfies the following equation:

$$Mel(f[i+1]) - Mel(f[i]) = Mel(f[i]) - Mel(f[i-1]). \tag{10}$$

Step 4: The Mel spectrum obtained in the previous step is transformed into the time domain, and the result is the MFCCs. Because MFCCs are real numbers, they can be transformed into the time domain using a DCT. The MFCC equation is as follows.

$$c_i = \sqrt{\frac{2}{N}} \sum_{j=1}^{P} m_j cos\left[\frac{\pi i}{P}(j - 0.5)\right] \tag{11}$$

The normalized energy was calculated as the 13th dimensional component of the feature vector for all scoring frames in the speech signal or speech signal file. In this study, 13 dimensions were considered, and the average value of all frames in each dimension was considered as the acoustic feature.

## 4. Experimental Setup and Procedure

The main automated hammering device used in this study included a voltage-controlled solenoid hammerhead that can automatically hammer at different frequencies, six-degree-of-freedom robotic arm to guide the hammerhead, and radio device consisting of a directional microphone, microphone power module, wireless transmission, and computer [32]. The automatic hammering device ensures that the hammering position and hammering acoustic data obtained are sufficiently standardized, and that the hammering acoustic data obtained by hammering at different frequencies can be used for machine learning, as shown in Figure 6.

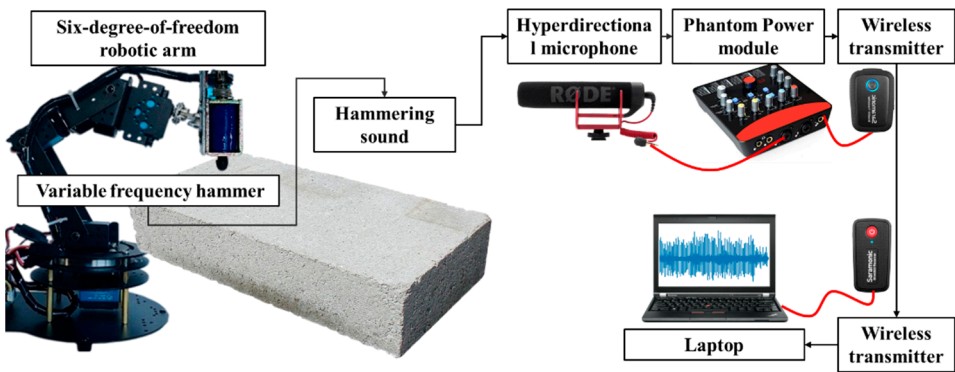

**Figure 6.** Schematic of the experimental setup.

Concrete specimens with a length of 420 mm, width of 60 mm, and height of 60 mm were used in this experiment. The concrete specimen was bent through a four-point bending system, as shown in Figure 7, producing damage distributed over the entire specimen as in Figures 8 and 9. In this study, the formation of damage classification is divided into two types: two-category, as shown in Figure 8, and four-category, as shown in Figure 9. The entire sample was divided into 21 groups. In the two-category classification, the concrete samples were divided into healthy and damaged areas. The healthy area was not damaged within 20 mm, and the damaged area included surface and internal damages. In the four-category classification, the samples were divided into healthy areas with no damage within 20 mm, surface damage areas with only surface cracks within 20 mm, internal damage areas with only internal cracks within 20 mm, and compound damage areas with both surface and internal cracks within 20 mm. Acoustic data were obtained by hammering the concrete from above at a hammering frequency of two times per second (2 Hz), five times per second (5 Hz), and 10 times per second (10 Hz). In the two-category and four-category classification, the concrete was subjected to 1260 hammer blows at 2 Hz, 3150 hammer blows at 5 Hz, and 6300 hammer blows at 10 Hz. The sound acquisition device is directional, pointing vertically in the hammering direction and moving with the hammering to ensure that the sound propagation distance does not need to be corrected.

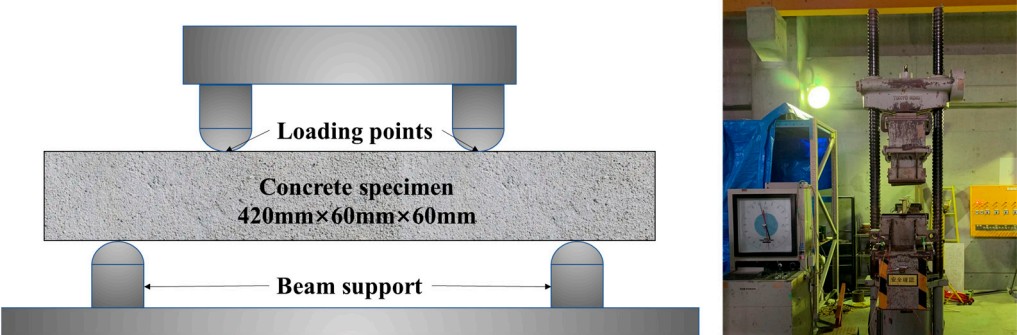

**Figure 7.** Schematic diagram of concrete specimen and four-point bending test.

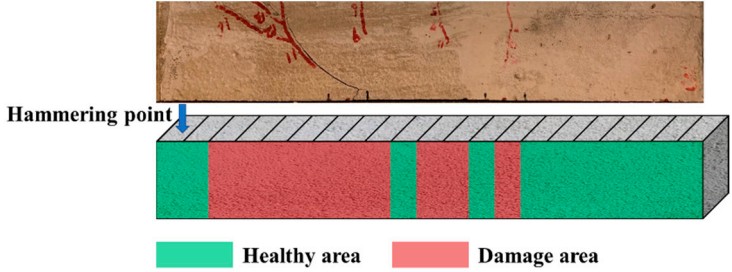

**Figure 8.** Schematic diagram of the concrete damage area for the two-category classification.

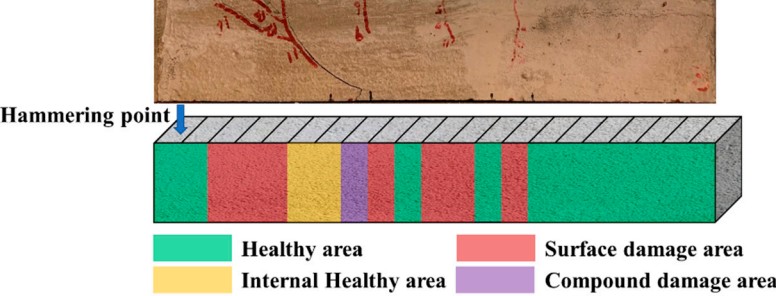

**Figure 9.** Schematic diagram of the concrete damage area for the four-category classification.

In this study, ReliefF was used for feature selection, and SVM was used to classify the damage types of concrete. Linear, quadratic, and Gaussian functions were used as kernel functions to build the model. In this study, acoustic features extracted from the selected signals based on two different hammering methods (single-frequency hammering and variable-frequency hammering) were used for classification and comparison.

## 5. Experimental Results and Discussion

### 5.1. Power Spectral Density Analysis

The sound signals recorded during the experiment were used for the PSD analysis. In this study, an automatic hammering device was used to hammer the area at three preset hammering frequencies under four different conditions for a total of 12 datasets. A typical acoustic wave was selected for each dataset. It was used to observe changes in the distribution and magnitude of the acoustic wave power at different hammering frequencies. The acoustic differences between variable-frequency hammering and single-frequency hammering were compared.

The PSD of the selected signal under different conditions was obtained for the frequency-domain analysis. As shown in Figure 10, the power was concentrated at 0–2000 Hz for the healthy category shown in Condition 1. Under different hammering frequencies, the power distribution did not change significantly with the hammering frequencies. In Condition 2, the shallow cracks (surface damage) were concentrated in

the range of 0–4000 Hz. Under different hammering frequencies, the distribution of the power size with hammering frequency change was not obvious. At the crack extension of Condition 3 (internal damage), the power was concentrated at 0–3000 Hz and 4000–5000 Hz, and the variation in power magnitude was evident under different hammering frequencies. At the compound damage of Condition 4 (internal damage + surface damage), the power was concentrated at 0–3000 Hz, and the variation in power magnitude was evident under different hammering frequencies. Four different types of damage with different power distributions in the same component can be easily distinguished by vertical observation. However, for Condition 3 at crack propagation and Condition 4 at composite damage, the power of these two adjacent damages under the same fixed hammering is concentrated in the same frequency distribution. As a result of using variable-frequency hammering, the power change in Condition 3 in the frequency distribution was significantly different from that of Condition 4. The change in Condition 3 was significant at 0–1000 Hz, and that of Condition 4 was significant at 2000–4000 Hz. The frequency distribution of the power changes in Condition 3 was significantly different from that of those in Condition 4. Condition 3 showed a significant change at 0–1000 Hz, while Condition 4 showed a significant change at 2000–4000 Hz. With single-frequency hammering, the difference was more pronounced for the two categories of health and damage. However, acoustic signals obtained by hammering at different frequencies can differentiate between two or more similar damage classifications.

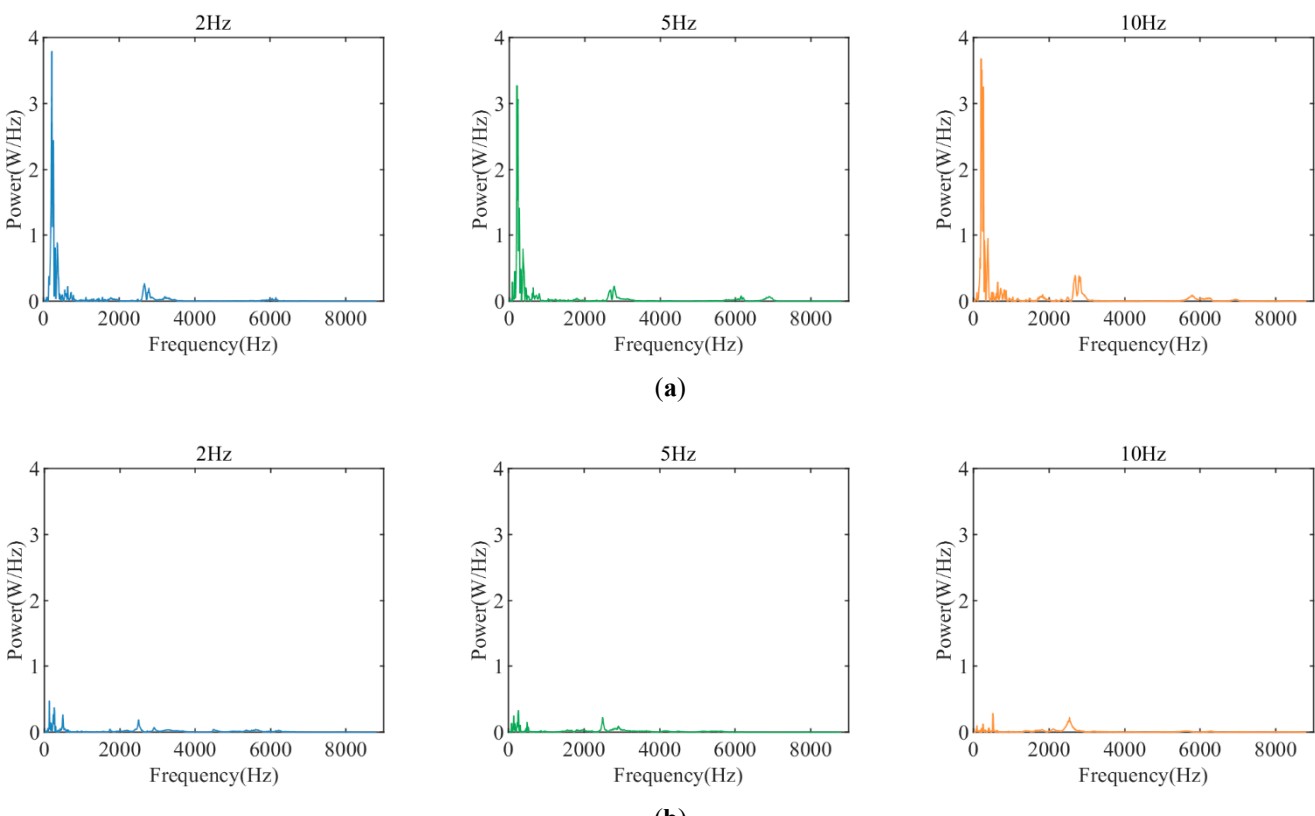

**Figure 10.** *Cont.*

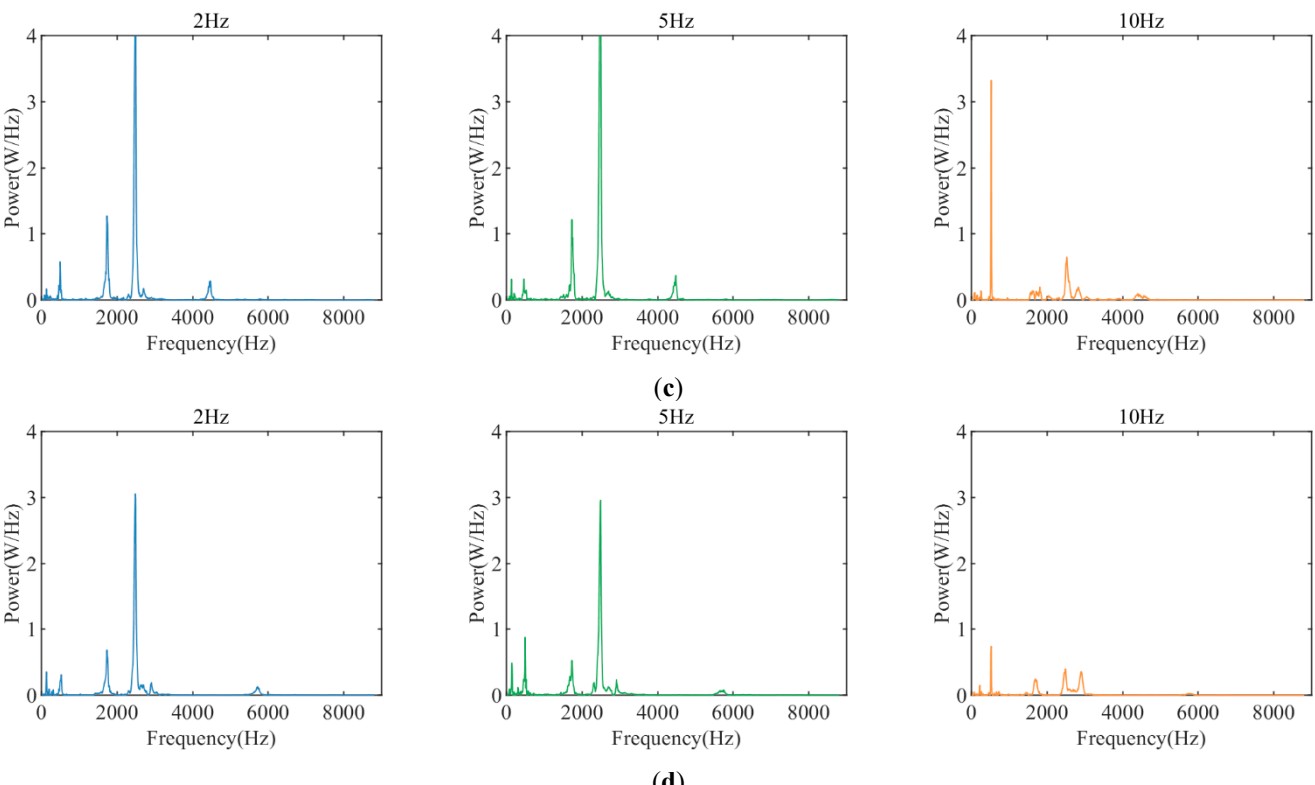

**Figure 10.** Twelve typical acoustic analyses of four-category under three frequency hammerings (**a**) Condition 1: Healthy area; (**b**) Condition 2: Surface damage area; (**c**) Condition 3: Internal damage area; (**d**) Condition 4: Compound damage area.

The experimental results demonstrated that variable-frequency hammering contains more information than single-frequency hammering, particularly for compound damage with two types of damage within 20 mm, which is challenging to identify using single-frequency hammering alone. In actual detection, the hammering frequency is not guaranteed to correspond to the hammering frequency that can be effectively identified; therefore, variable-frequency hammering is required.

*5.2. ReliefF-Based Feature Selection*

According to the ReliefF evaluation, a sequence of features sorted by correlation from largest to smallest can be obtained. It is only necessary to decide how many of the features with the lowest ranking can be deleted to perform feature selection. In this study, two types of classification were used for damage; thus, the selection of acoustic features obtained from frequency hammering was also divided into two types: feature selection based on two classifications (Table 1) and feature selection based on four classifications (Table 2). These were added together with the MFCCs of single-frequency hammering using SVM for comparison.

**Table 1.** First 13 dimensional features in two-category classification.

| Features | ReliefF |
|---|---|
| The 3rd dimension of 5 Hz | 0.0659 |
| The 1st dimension of 5 Hz | 0.0574 |
| The 6th dimension of 5 Hz | 0.0567 |
| The 12th dimension of 5 Hz | 0.0530 |
| The 5th dimension of 2 Hz | 0.0527 |
| The 10th dimension of 5 Hz | 0.0519 |
| The 2nd dimension of 5 Hz | 0.0478 |
| The 11th dimension of 5 Hz | 0.0477 |
| The 9th dimension of 5 Hz | 0.0477 |
| The 4th dimension of 5 Hz | 0.0476 |
| The 13th dimension of 5 Hz | 0.0471 |
| The 7th dimension of 5 Hz | 0.0463 |
| The 11th dimension of 10 Hz | 0.0446 |

**Table 2.** First 13 dimensional features in four-category classification.

| Features | ReliefF |
|---|---|
| The 3rd dimension of 5 Hz | 0.0813 |
| The 4th dimension of 5 Hz | 0.0795 |
| The 5th dimension of 2 Hz | 0.0772 |
| The 1st dimension of 5 Hz | 0.0669 |
| The 12th dimension of 5 Hz | 0.0645 |
| The 7th dimension of 5 Hz | 0.0623 |
| The 6th dimension of 5 Hz | 0.0601 |
| The 9th dimension of 5 Hz | 0.0587 |
| The 3rd dimension of 10 Hz | 0.0586 |
| The 11th dimension of 5 Hz | 0.0569 |
| The 8th dimension of 5 Hz | 0.0557 |
| The 13th dimension of 5 Hz | 0.0548 |
| The 11th dimension of 10 Hz | 0.0527 |

In addition, the experimental results of feature selection showed that for the damage selected in this study, feature selection was more inclined toward a specific hammering frequency. In addition, based on the number of classifications, the required hammering frequency changed after the addition of the surface, internal, and combined surface-internal damage classifications, except for the offset single hammering frequency. Different instances of damage have different degrees of sensitivity corresponding to different hammering frequencies. Exploring this degree of sensitivity can help to further classify concrete damage.

### 5.3. Comparison between Single-Frequency Hammering and Variable-Frequency Hammering

In this study, the vector data of the single-frequency hammer MFCCs and variable-frequency hammer MFCCs obtained from two and four categories were imported into the SVM model, which used linear, quadratic, and Gaussian kernel functions and a one-to-one multi-class classification strategy. For accuracy testing, the six-fold cross-validation method was used to divide the dataset into six parts, five of which were used for training and one for testing, and the mean of the results of the six times was used as an estimate of the accuracy of the algorithm.

### 5.3.1. Classification Using Single-Frequency Hammering MFCCs

Tables 3 and 4 show the SVM classification results of the single-frequency hammering MFCCs in the two- and four-classification cases under the three kernel functions. In the case of two classes, the accuracy of each single-frequency hammering can exceed 90%. In the case of the four-category classification, the accuracy of the three single-hammer frequencies

decreased slightly compared with that of the two-level classification. As observed, the accuracy at 2 Hz is lower than that at 5 Hz and 10 Hz; this indicates that MFCCs are effective as an acoustic feature for classification, whereas the sensitivity of the hammering frequency varies for different forms of damage classification.

**Table 3.** Accuracy of each single-frequency hammering in the two-category classification.

| Kernel Type | 2 Hz | 5 Hz | 10 Hz |
|---|---|---|---|
| Linear | 84.1% | 82.% | 83.1% |
| Quadratic | 91.7% | 94.0% | 90.8% |
| Gaussian | 86.3% | 96.8% | 86.3% |

**Table 4.** Accuracy of each single-frequency hammering in the four-category classification.

| Kernel Type | 2 Hz | 5 Hz | 10 Hz |
|---|---|---|---|
| Linear | 77.1% | 82.2% | 79.4% |
| Quadratic | 88.9% | 93.0% | 91.1% |
| Gaussian | 85.7% | 96.5% | 85.7% |

5.3.2. Classification Using Variable-Frequency Hammering MFCCs

Table 5 shows the SVM classification results for two classification forms of the variable-frequency hammering MFCCs under three kernel functions, with the highest accuracy of over 97% for both classification forms, which is higher than that of the three single-frequency hammering cases. This demonstrates that the frequency conversion of MFCCs is effective at classifying concrete damage and better than the single-frequency MFCC, that is, compared with single-frequency hammering, the acoustic characteristics obtained from the acoustic signals of variable-frequency hammering are more effective at classifying the damage above the second category in Section 5.3.1.

**Table 5.** Accuracy of each variable-frequency hammering in the two-category and four-category classifications.

| Kernel Type | Two Categories | Four Categories |
|---|---|---|
| Linear | 84.1% | 89.2% |
| Quadratic | 94.0% | 96.2% |
| Gaussian | 97.5% | 97.8% |

The experimental results showed that the classification prediction accuracy of the method based on variable-frequency hammering for concrete is high, reaching 97.5% for the two categories and 97.8% for the four categories. Compared with single-frequency hammering, it is able to distinguish multiple types of damage, surface, internal, and combined surface–internal, more effectively.

## 6. Conclusions and Future Work

In this study, valid information was obtained without increasing dimensionality to accurately classify multiple types of damage occurring in the same concrete component. A detection method based on the acoustic features of variable-frequency hammering was proposed. Variable-frequency hammering MFCCs are generated from acoustic data obtained by variable-frequency hammering through feature selection and extraction; they are used as the main feature of SVM classification for multi-category damage classification.

The acoustic features obtained by single-frequency hammering were used as classification features and compared with those generated by variable-frequency hammering. The results showed that the SVM models based on both single-frequency hammering and variable-frequency hammering can accurately detect both two- and four-category concrete damage occurring in the same specimen. However, the classification model based on variable-frequency hammering showed a higher accuracy of 97.8% than that based on

single-frequency hammering, which indicates that variable-frequency hammering is more suitable as a feature for multi-category damage classification than single-frequency hammering. In addition, the experimental results of PSD analysis and feature selection showed that variable-frequency hammering contains more effective information than single-frequency hammering. Furthermore, variable-frequency hammering can compensate for the lack of information of single hammering from common single-frequency hammering methods. Variable-frequency hammering can be combined with machine learning to detect concrete damage more accurately without causing a dimensional disaster. Simultaneously, the experimental results proved that different types of damage have different sensitivities to different hammering frequencies. However, the tests are still at the research laboratory stage and may not solve existing engineering problems in the field, such as detecting damage in noisy and topographically complex environments. In the future, we will improve the detection method such that hammering and signal processing can be performed automatically using hammering equipment and micro-phones, and the hammering frequency can be adjusted intelligently for different dam-ages to develop an intelligent frequency-hammering robot with great potential in the field of inspection.

**Author Contributions:** Conceptualization, H.H.; Methodology, X.H.; Validation, X.H.; Formal analysis, X.H.; Investigation, X.H.; Data curation, X.H.; Writing—original draft, X.H.; Writing—review & editing, H.H. and Z.W.; Supervision, H.H. and Z.W.; Project administration, Z.W. All authors have read and agreed to the published version of the manuscript.

**Funding:** This research was funded by the National Key Research and Development Program of China (Grant number: 2019YFC1511103).

**Conflicts of Interest:** The authors declare no conflict of interest.

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
