# Peer review of "Development of a Variable-Frequency Hammering Method Using Acoustic Features for Damage-Type Identification"

_applsci, doi:10.3390/app13031329_

Round 1

Reviewer 1 Report

Dear Authors, congratulations on the research carried out. The paper entitled "Development of a variable-frequency hammering method using acoustic features for damage-type identification" is well structured and concisely shows the research carried out. However, I think that general sections such as section 3 should be reduced.

Particular Comments,

- In section 3.1, line 114, the citation of the work of Kira et al. (1992) does not appear in references. Please add it and correct the numbering of all of them.

- In section 3.2, line 146, review the quote from Vapnik et al., it should put, Vapnik et al. (2013).

- In section 5.3.2, lines 358-362, the results should be compared with those of other authors using simple hammering. In addition, the other methods should be compared in time and price.

- On the other hand, it is not specified whether this test can be carried out in situ, since depending on how it can be extracted from the article, a large number of very specific devices and a robotic arm are needed. Please mention if it can be done.

Sincerely,

Author Response

The number of lines in the manuscript has changed, so we have revised the response document, please refer to the latest document.

Reviewer 2 Report

The authors proposed a variable-frequency features-based ML model to identify damage type in concrete beam type of structure. However, the area of research is very prominent and is the need of the hour. However, the reviewer has the following concerns about the presented manuscript. 

1. The article explicitly mentions 'damage type'; however, identifying damage type has multiple connotations. Do the authors mean different kinds of damages or different damages with the variable extent of damage. Please clarify it the paper body and explain it in your explanation of this review. 

2. ML-based research is booming for various civil engineering/SHM/damage identification, however, are they the best solution for the problem is doubtful. Why did the authors use MFCC coefficients with SVM and not random forest or XGBoost? Please clarify

3.  Can your proposed algorithm detect damages due to Delayed Ettringite formation and Alkali silica reaction, as they have similar phenomena of creating damage in concrete structures? If you are not aware of these phenomena, please read and provide your explanation.

4. The introduction and literature of this manuscript is wholly based on the area of author's work for this paper, It is suggested to add a variety of the past work working on damage detection using various fields such as Time-frequency analysis, time-series forecasting, ML/DL work to establish a diverse point of view in the introduction of this paper. Following are some of the examples to cite in this article

(a) Gillich, G.-R., & Praisach, Z.-I. (2014). Modal identification and damage detection in beam-like structures using the power spectrum and time–frequency analysis. Signal Process., 96, 29–44. 

(b) Limongelli, M. P., Manoach, E., Quqa, S., Giordano, P. F., Bhowmik, B., Pakrashi, V., & Cigada, A. (2021). Vibration Response-Based Damage Detection. Structural Health Monitoring Damage Detection Systems for Aerospace. Springer.

(c) Sony, S and Sadhu, A. (2019). Identification of progressive damage in structures using time-frequency analysis, CSCE General Conference, Montreal, Canada.

(d) Avci, O., Abdeljaber, O., Kiranyaz, S., Hussein, M., Gabbouj, M., & Inman, D. J. (2021). A review of vibration-based damage detection in civil structures: From traditional methods to Machine Learning and Deep Learning applications. Mech. Syst. Sig. Process., 147, 107077.

(e) Rastin, Z., Ghodrati Amiri, G., & Darvishan, E. (2021). Unsupervised Structural Damage Detection Technique Based on a Deep Convolutional Autoencoder. Shock Vib., 2021.

Reviewer 3 Report

The article is interesting. A detection method based on the acoustic features of variable-frequency hammering is proposed. However, the reviewer has the following comments:

1. I propose to change the names of sub-points in chapter 3. 3.1. Algorithm Relief, 3.2. Support vector machine method, 3.3. Mel-frequency cepstrum coefficients method

2. Figure 4 should be a table or text, not a picture

3. What do the numbers on the beams in Fig. 9 and 10 mean? Load stage? Load size?

4. On line 244 it says: “Damage distributed throughout the specimen was produced by loading the stress points, as shown in Figure 8”. Meanwhile, Fig. 6 shows only the view of the beam and load forces.

5. Please write that the beam was tested in a four-point bending system

6. How many samples were tested - only one?

7. The conclusions are pretty obvious. The article would be more interesting if more elements with different geometries and made of different concretes were examined. Nevertheless, the reviewer accepts the paper for publication after making the corrections indicated above.

Reviewer 4 Report

Dear Authors

In this study, valid information was obtained without increasing dimensionality to accurately classify multiple types of damage occurring in the same concrete component.  A detection method based on the acoustic features of variable-frequency hammering is proposed. Variable-frequency hammering MFCCs are generated from acoustic data obtained by variable-frequency hammering through feature selection and extraction; they are used as the main feature of SVM classification for multi-category damage classification. 

The acoustic features obtained by single-frequency hammering were used as classification features and compared with those generated by variable-frequency hammering. The results show that both the SVM model based on a single-frequency hammering and the variable-frequency hammering can accurately detect both two- and four-category concrete damages occurring in the same specimen.

 To make the discussion more homogeneous and complete I suggest the following bibliography:

1. Fabbrocino F., Farina I., Modano M., Loading noise effects on the system identification of composite structures by dynamic tests with vibrodyne, COMPOSITES. PART B, ENGINEERING (IF: 6.864), 2017, Vol. 115, Pag. 376-383, ISSN: 1359-8368, DOI: 10.1016/j.compositesb.2016.09.032;

2. Funari M.F., Spadea S., Lonetti P., Fabbrocino F., Luciano R., Visual programming for structural assessment of out-of-plane mechanisms in historic masonry structures, April 2020, Journal of Building Engineering (IF: 5.318) 31(4–5):101425, DOI: 10.1016/j.jobe.2020.101425;

3. Modano, M., Fabbrocino, F., Gesualdo, A., Matrone, G., Farina, I.,  Fraternali, F. On the forced vibration test by vibrodyne, COMPDYN 2015 - 5th ECCOMAS Thematic Conference on Computational Methods in Structural Dynamics and Earthquake Engineering, 209-217, ISSN   00002015.

4. Carpentieri, G., Modano, M., Fabbrocino, F., Feo, L., Fraternali, F., On the minimal mass reinforced of masonry structures with arbitrary shape, MECCANICA (IF: 2.316), (2017)  Vol. 52, Issue 7, Pag. 1561-1576, ISSN: 0025-6455, DOI: 10.1007/s11012-016-0493-0.

Best regards

Round 2

Reviewer 2 Report

Authors have improved the manuscript